# An Empirical Study of Reasoning Steps in Thinking Code LLMs

Haoran Xue
York University
North York, Canada
hrx00@yorku.ca

Gias Uddin
York University
North York, Canada
guddin@yorku.ca

Song Wang
York University
North York, Canada
wangsong@yorku.ca

## Abstract

Thinking Large Language Models (LLMs) generate explicit intermediate reasoning traces before final answers, potentially improving transparency, interpretability, and solution accuracy for code generation. However, it remains unclear which trace characteristics are informative and the quality of the reasoning chains. In this paper, we present an empirical study examining the reasoning processes and the quality of thinking LLMs on code generation tasks. We evaluate six state-of-the-art reasoning LLMs (DeepSeek-R1, OpenAI-o3-mini, Claude-3.7-Sonnet, Gemini-2.0-Flash-Thinking, Gemini-2.5-Flash, and Qwen-QwQ) on 100 BigCodeBench code generation tasks (600 model–task instances; 3,772 reasoning steps). To characterize reasoning-chain structure, we measure step count and per-step verbosity, and compare successful versus failed attempts under difficulty stratification (*Hard* vs. *Non-Hard*). We further perform a 21-participant human evaluation of reasoning quality across three dimensions: efficiency, logical consistency, and completeness, and we build a taxonomy of problematic reasoning patterns. We find the model- and difficulty-dependent relationship between step count and success, and verbosity is not a reliable correctness signal. Human analysis indicates that completeness issues dominate failures (44.5%), most often due to missed edge cases and boundary conditions, and incompleteness is a stronger predictor of failure on *Hard* tasks than on *Non-Hard* tasks ($\rho = -0.219$ vs. $\rho = -0.096$).

## CCS Concepts

• **Software and its engineering** → **Source code generation**; **Empirical software validation**; • **Computing methodologies** → **Natural language generation**.

## Keywords

Large Language Models, Large Reasoning Models, Code Generation

**ACM Reference Format:**
Haoran Xue, Gias Uddin, and Song Wang. 2026. An Empirical Study of Reasoning Steps in Thinking Code LLMs. In *Proceedings of the 3rd ACM International Conference on AI-Powered Software (AIware '26), July 6–7, 2026, Montreal, QC, Canada.* ACM, New York, NY, USA, 10 pages. https://doi.org/10.1145/3805760.3814905

## 1 Introduction

LLMs have made major progress on software engineering tasks such as code generation [19, 29], program translation [3, 17, 26], and bug repair [14, 40]. Despite strong pass@1 accuracy on many benchmarks [2, 6], conventional (non-thinking) LLMs still struggle on complex coding tasks where developers need transparent and controllable problem solving. Their intermediate hypotheses and decision points remain hidden, limiting independent verification and process-level auditing [36]. Even with chain-of-thought prompting, the model's written rationale may not reflect its actual decision process [36]. Multi-step programming (decomposition, edge cases, coordinating APIs/state/constraints) is also fragile and prompt-sensitive, and failures provide little process signal for localizing breakdowns and debugging [24, 25].

Recent thinking LLMs aim to address this by externalizing intermediate reasoning traces before producing final solutions [39]. These models generate step-by-step reasoning that states problem understanding, strategy, implementation choices, constraints, and edge cases (e.g., OpenAI's o1/o3 series [30, 31], DeepSeek-R1 [12], and Claude reasoning variants [1]). In principle, explicit reasoning could improve interpretability and trust and help produce correct code with explanations [21]. However, the quality of these reasoning traces remains underexplored. Most evaluations of coding LLMs emphasize final correctness (e.g., pass@k) [2, 6, 18], which revelas outcome differences but does not indicate whether the reasoning process is effective or where it fails. This process-level gap is significant because developers are increasingly relying on reasoning traces to validate, debug, and audit model-produced code.

We address this gap with a process-centered empirical study that connects reasoning-chain structure and quality to solution outcomes across thinking code LLMs. We evaluate six state-of-the-art reasoning LLMs, DeepSeek-R1, OpenAI-o3-mini, Claude-3.7-Sonnet, Gemini-2.0-Flash-Thinking, Gemini-2.5-Flash, and Qwen-QwQ, on 100 tasks sampled from BigCodeBench-Instruct [13]. These models were selected because, at the time of data collection, they were widely used "thinking" code LLMs with publicly identifiable endpoints and official APIs, spanning both open and closed ecosystems to reflect the range of reasoning behaviors developers can deploy. We ask:

**RQ1.** Do the length and depth of the reasoning traces, operationalized as (i) step count and (ii) per-step verbosity, relate to the generation of correct solutions? (**Reasoning Process Analysis.**)

**RQ2.** How do problematic reasoning steps contribute to the generated incorrect solutions? (**Reasoning Quality Analysis.**)

Methodologically, we generate 600 solutions (6 models × 100 tasks) reporting results separately for *Hard* and *Non-Hard* tasks. In BigCodeBench, *Hard* tasks require more than two libraries, have longer-than-average reference solutions, and have solve rates below 50% across evaluated models, while the remaining tasks are treated as *Non-Hard* (our sample contains 14 *Hard* and 86 *Non-Hard* tasks), and record 3,772 reasoning steps. For **RQ1**, we answer it

through two sub-questions: RQ1.1 analyzes whether step counts differ between successful and failed cases, and RQ1.2 analyzes whether verbosity (words per step) is linked to success. For **RQ2**, we conduct a user study with 21 graduate students labeling reasoning steps on three defined criteria inspired by the literature, efficiency [41], logical consistency [23], and completeness [4], which capture complementary aspects of reasoning chains. We then identify and classify the problematic patterns and the root causes, and derive a taxonomy across the three dimensions.

We find that the step–count–success relationship is model- and difficulty-dependent: on *Hard* tasks, some models benefit from longer chains (e.g., Gemini-2.0-FT, $\rho = 0.79$, 95% CI [0.461, 0.891]), while others (e.g., DeepSeek-R1) succeed with fewer steps, and on the *Non-Hard* split, correlations largely collapse toward zero. Verbosity per step varies widely yet does not reliably predict correctness. Human evaluation identifies completeness as the dominant failure mode (44.5%), driven mainly by missing edge-case handling (32.17%), and more strongly associated with failure on *Hard* tasks. In summary, our study offers the following contributions:

- We present an empirical study of six state-of-the-art thinking code LLMs on 100 BigCodeBench-Instruct tasks, varying *Hard* and *Non-Hard* difficult levels, focusing on the reasoning process and quality analysis.
- We conduct a human study with 21 graduate students to assess reasoning quality along with three dimensions, efficiency, logical consistency, and completeness.
- We analyze the reasoning traces labelled by participants and develop a taxonomy of problematic reasoning patterns and their root causes.

**Data and code:** https://github.com/xhinini/Reasoning-LLMs/tree/main.

## 2 Related Work

Recent work on LLM-based code generation reports striking benchmark gains, but also highlights evaluation gaps across the software development lifecycle. Surveys show pass@1 rising from early single-digit performance (e.g., 3.6%) to state-of-the-art results exceeding 90%, with instruction-tuned models consistently outperforming base models and the open-/closed-source gap narrowing quickly [19]. LLMs are now widely used for description-to-code translation, code completion, and program repair, and studies of tools such as ChatGPT suggest meaningful productivity benefits—though results remain highly tool- and task-dependent [37]. However, prevailing evaluations still emphasize functional correctness and security, with limited attention to maintainability, readability, and human-facing qualities [37]. This limitation is pronounced for class-level generation: ClassEval reports that even GPT-4 achieves only 37.0% correctness on interdependent class methods, compared to 85.4% on simpler function-level tasks [9]. Together, these findings suggest that high benchmark scores can mask brittle behaviours and motivate more comprehensive assessments of the trustworthiness and reliability of generated code.

In parallel, reasoning-capable LLMs are increasingly central to software engineering automation. Recent surveys and empirical studies (e.g., 123 papers) group approaches into Code Chain-of-Thought variants, execution-based reasoning with runtime feedback, inference scaling via sampling/search, and agentic multi-agent

workflows [5, 42, 44]. Methods such as SWE-RL further apply reinforcement learning to improve reasoning for real-world SE tasks, reporting notable progress on SWE-bench Verified [20, 38], and the field is shifting from single-shot generation toward pipelines evaluated across localization, bug fixing, test generation, and software analytics [15]. Yet, evidence suggests that LLMs still struggle more with semantic-heavy tasks (e.g., vulnerability detection) than with syntax-oriented tasks (e.g., summarisation, repair) [44], supporting a view of LLMs as assistants rather than substitutes for professional developers. Similar brittleness appears beyond SE: medical LLMs can achieve high diagnostic accuracy while omitting essential reasoning steps [32] and may exhibit clinically concerning biases despite strong benchmark scores [28]; in mathematics, step-level errors persist even when final answers are correct [43], and performance can collapse under superficial perturbations or irrelevant information [27]. These results motivate systematic, fine-grained evaluations of reasoning quality, rather than relying solely on final correctness, to better assess the trustworthiness of generated code.

## 3 Study Setup

### 3.1 Studied Code Generation Tasks

We select *BigCodeBench* [13], a function-level code-generation benchmark, for two reasons. First, it offers broad, real-world coverage, with 1,140 Python tasks drawn from 139 popular libraries across seven domains (computation, visualization, general utilities, time operations, system operations, networking, and cryptography). The benchmark contains about 5.6 test cases per task on average and reports roughly 99% average test coverage. It also includes 148 filtered more difficult "*Hard*" tasks (13.16%) defined by requiring more than 2 libraries, having solutions longer than the average length (426 tokens), and having solve rates below 50% across evaluated models, plus 992 remaining "*Non-Hard*" tasks (87%) [13]. Second, unlike short, self-contained algorithmic benchmarks such as *HumanEval* [6] and *MBPP* [2], it targets practical instruction-following and multi-library integration, which is necessary to test the limits of thinking code LLMs and expose problematic reasoning steps. *BigCodeBench* provides two evaluation splits: *BigCodeBench-Complete*, which uses structured docstrings with comprehensive technical details, and *BigCodeBench-Instruct*, which presents concise natural-language instructions closer to typical user requirements [13]. In each task, the model is asked to implement a single target function, and correctness is evaluated using associated unittest-style test code through the benchmark's evaluator backend.

From the 1,140 tasks, we randomly sampled 100 tasks from *BigCodeBench-Instruct* split using standard statistical sampling (95% confidence level, 9-10% margin of error), a methodology widely used in software engineering empirical studies [35]. We refer to the entire benchmark (Hard + non-Hard) as the *full benchmark*, and we refer to the remaining tasks after removing *Hard* as *Non-Hard* for clarity. The sample contains 14 *Hard* tasks (14%) and 86 *Non-Hard* tasks (86%), closely matching the dataset distribution.

### 3.2 Studied Reasoning LLMs

We study six thinking LLMs, i.e., DeepSeek-R1 [12], OpenAI-o3-mini [31], Claude-3.7-Sonnet [1], Gemini-2.0-Flash-Thinking [11],

**Table 1: Studied reasoning ("thinking") LLMs. Dates/params shown only when verified; otherwise "—".**

| Model | Date | Provider | Params | Ref |
|---|---|---|---|---|
| DeepSeek-R1 | 2025.05 | DeepSeek | 671B | [7, 10, 12] |
| OpenAI o3-mini | 2025.01 | OpenAI | — | [31] |
| Claude-3.7-Sonnet | 2025.02 | Anthropic | — | [1] |
| Gemini-2.0-Flash-Thinking | 2025.01 | Google | — | [11] |
| Gemini-2.5-Flash | 2025.05 | Google | — | [16] |
| Qwen-QwQ | 2024.11 | Qwen (Alibaba) | 32B | [43] |

**Table 2: Summary of reasoning data collected for manual evaluation. Each model was evaluated on 100 tasks (600 total).**

| LLM Model | Steps | Avg. |
|---|---|---|
| OpenAI-O3-mini | 585 | 5.85 |
| Claude-3.7-Sonnet | 570 | 5.70 |
| Qwen-QwQ | 502 | 5.02 |
| DeepSeek-R1 | 558 | 5.58 |
| Gemini-2.5-Flash | 737 | 7.37 |
| Gemini-2.0-Flash-Thinking | 820 | 8.20 |
| **Total** | **3,772** | **6.29** |

Gemini-2.5-Flash [16], and Qwen-QwQ [43], spanning different sizes, versions, and release periods (Table 1), covering both open- and closed-source models; all models were accessed via their official APIs.

## 3.3 Data Preparation

We begin by running each of the six reasoning LLMs on our 100-task sample and storing the reasoning traces and code solutions. In total, we collected 3,772 reasoning steps from 600 task instances (Table 2), averaging 6.29 steps per task. We store all outputs as structured JSON files (one per task per model). Notably, models vary in how they expose or format reasoning traces. For example, DeepSeek-R1 provides a built-in separation between the reasoning process and the final response (e.g., "content" vs. "reasoning_content"), allowing direct access to raw reasoning [8], whereas Gemini-2.0-Flash-Thinking and Qwen-QwQ embed reasoning and answers in a single output, which is also reported in a prior observation [32]. Because providers expose reasoning in different surface formats, we elicit a single standardized, user-visible, step-delimited rationale while preserving the substance of each model's response, so step and verbosity analyses reflect the model's expressed intermediate steps rather than any unobserved internal computation. Since raw traces are often lengthy and hard to interpret for human evaluation, we use a structured prompt to present reasoning in clearly delimited steps (e.g., <step1>, <step2>) across all models under comparable evaluation conditions. We report pass rates under a single fixed prompting and trace-formatting protocol shared by all models to enable controlled comparisons, so these numbers should be read as prompt-conditional performance rather than leaderboard-optimized upper bounds. This design improves comparability across models, but it may also introduce prompt-format effects, since different models can respond differently to the same reasoning template [33].

To support RQ2's human evaluation, we convert the JSON into human-readable text files, each containing exactly 10 tasks and the reasoning traces from all six models (60 reasoning sequences per file). Within each file, we organize content by task (problem description followed by labeled traces such as model1_reasoning_contents, etc.). We produce three identical copies of the full materials so that three evaluators can independently assess each task to improve reliability and reduce individual bias.

## 3.4 Evaluation Methodology

To assess reasoning quality from developers' perspectives, we conduct a human evaluation. Figure 1 summarizes the workflow across four phases: (1) Participant Recruitment, (2) Training Session, (3) Task Assignment, and (4) Evaluation Phase. We involve 21 participants with computer science backgrounds and familiarity with Python programming and LLMs. Before training, three researchers developed a coding guide through multiple rounds of discussion and refinement with concrete examples to ensure consistent evaluation procedures and operational definitions for three criteria:

- **Efficiency:** Measures whether each reasoning step meaningfully advances problem solving and identifies redundant steps. Participants count the total number of reasoning steps and label each step as "Essential" if the step introduces necessary information, performs required calculations, or makes crucial connections needed to solve the problem; or "Non-Essential", where the redundant or unnecessary elaboration does not advance the solution.
- **Logic Consistency:** Assesses coherence and internal consistency. For each step (except the first), participants judge whether it logically follows from previous steps, marking it as "Coherent" if the connection is clear and valid, or "Incoherent" if there are gaps or inconsistencies.
- **Completeness:** Checks whether the reasoning addresses all problem requirements. Participants identify all essential requirements in the problem statement, then mark each as "Addressed" or "Not addressed" in the reasoning. They also verify whether edge cases and constraints are considered and list any missed edge cases.

The three metrics together can capture complementary aspects of reasoning chains, enabling diagnosis beyond pass/fail. Participants then completed an approximately 10-minute training session covering study objectives, the three criteria, and annotated examples of reasoning traces. For task assignments, each participant evaluated 10 tasks sampled across six reasoning LLMs, and each task was rated by three participants to support reliability analysis. Participants then independently applied the guide to the assigned reasoning contents, spending about 2–2.5 hours per participant.

Because our downstream analyses operate on task-level continuous quality scores, we first aggregate each participant's step-level binary judgments into a single per-task ratio score for each criterion, we compute: **Efficiency Score** $= \frac{\text{Number of Essential Steps}}{\text{Total Number of Steps}}$, **Coherence Score** $= \frac{\text{Number of Coherent Steps}}{\text{Total Number of Steps - 1}}$, **Completeness Score** $= \frac{\text{Requirements Addressed}}{\text{Total Requirements}}$. This yields a task×participant score matrix with $k=3$ participant scores per task. We therefore quantify inter-rater agreement on these continuous per-task scores using the average-measures one-way random-effects intraclass correlation, $ICC(1, 3)$ [34], which matches our design where each task is rated by three participants randomly sampled from a larger participant pool. Formally, $\text{ICC}(1, k) = \frac{MS_B - MS_W}{MS_B}$, where $MS_B$ is the one-way

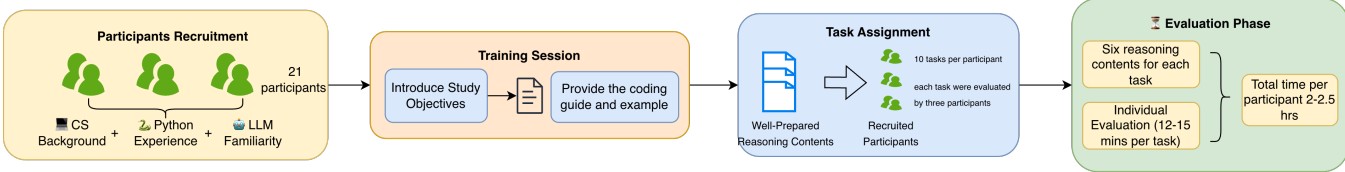

Figure 1: Participant Recruitment and Evaluation Process

ANOVA mean square between tasks (targets/rows) computed over the $k$ participant scores per task, and $MS_W$ is the within-task (residual/error) mean square across participants.

We interpret ICC using standard bands (excellent > 0.9, good 0.75–0.9, moderate 0.5–0.75, and poor < 0.5) [22], and observe strong reliability across all three criteria—Efficiency (0.926), Logic Consistency (0.934), and Completeness (0.863).

## 4 Results

### 4.1 Assessing the Reasoning Process in Thinking Code LLMs (RQ1)

To provide a foundational understanding of how thinking LLMs structure their reasoning processes, we study two complementary aspects of reasoning chain characteristics and their effects: (i) we examine the quantitative structure of reasoning through step counts, as the number of reasoning steps may reflect the complexity of a model's problem-solving approach and potentially correlate with success rate (RQ1.1); and (ii) we analyze reasoning per-step verbosity to see wthether a more detailed articulation of reasoning improves solution quality (RQ1.2).

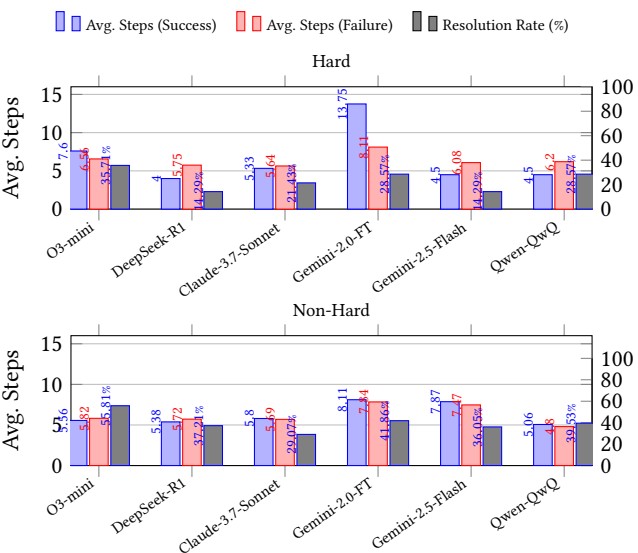

Figure 2: Average reasoning steps (success/failure) and resolution rate for each model, split by difficulty. Both left y-axes share the same range and tick interval (0, 5, 10, 15). Resolution rate uses the right y-axis.

#### 4.1.1 RQ1.1 Do the reasoning step counts differ between successful and failed cases?

*Approach.* By comparing step counts across successful and failed cases, we determined whether more elaborate reasoning processes

lead to better outcomes or whether failed attempts are characterized by either premature termination (i.e., too few steps) or excessive exploration (i.e., too many steps). First, we divide all six model runs on the same 100 tasks by the task difficulty splits (*Hard* vs. *Non-Hard*). Second, for each model and split, we log (i) the average number of steps on successes vs. failures, (ii) the overall resolution rate, and (iii) an association between step count and pass/fail using Spearman's rank correlation.

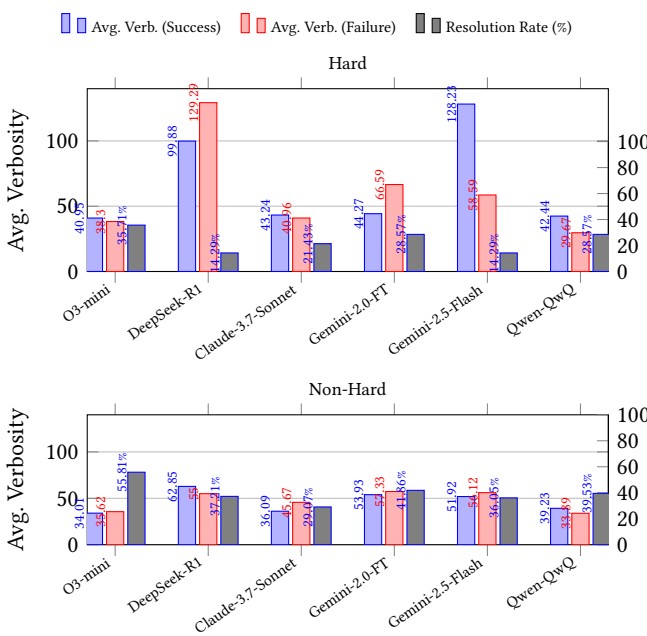

Figure 3: Average reasoning verbosity (success/failure) and resolution rate for each model, split by difficulty. Both left y-axes share the same range and tick interval (0, 50, 100). Resolution rate uses the right y-axis.

*Results.* As shown in Figure 2, on the *Hard* set, Gemini-2.0-FT demonstrates the clearest stepwise reasoning benefit: successful attempts average 13.75 steps vs. 8.10 on failures, with a strong positive correlation between step count and success (Spearman value $\rho = 0.79$, 95% CI [0.461, 0.891]). This model appears to benefit from extended exploration, in that the more steps it takes, the more likely it is to find the solution, achieving a 28.57% resolution rate on *Hard* set problems. O3-mini also shows a relatively high correlation (Spearman value $\rho = 0.219$). In contrast, DeepSeek-R1 (Spearman value $\rho = -0.579$, 95% CI [-0.837, -0.381]) and Qwen-QwQ show the opposite pattern that succeeds with fewer steps (Spearman value $\rho = -0.52$, 95% CI [-0.826, -0.143]). This model appears to benefit

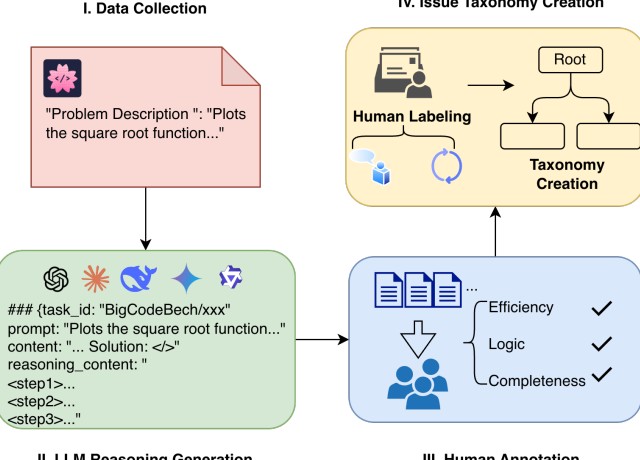

**Figure 4: Steps for evaluating the quality of reasoning contents**

more from conciseness, as longer chains may indicate that it is struggling or taking unproductive paths. Other models, such as Claude-3.7-Sonnet, show weak step-success correlations on *Hard* tasks, suggesting their performance is less dependent on chain length. Notably, when tasks become easier on the *Non-Hard* set, all step-success correlations collapse toward zero, and resolution rates improve across the board (29.1% to 55.8% compared to 14.3% to 35.7% on *Hard*). Furthermore, the average step counts between successful and failed attempts converge on *Non-Hard* tasks (differences of ≤1 step for most models), contrasting sharply with the divergent patterns on *Hard* tasks. The optimal chain length for *Hard* problems appears to be model-specific.

> **Answer to RQ1.1** Step count has model-specific, difficulty-dependent relationships with success. On *Hard* tasks, some models benefit from longer chains (Gemini-2.0-FT: $\rho = 0.79$; O3-mini: $\rho = 0.22$), while others succeed with shorter chains (DeepSeek-R1: $\rho = -0.58$; Qwen-QwQ: $\rho = -0.52$). On *Non-Hard* tasks, correlations approach zero, and success–failure step gaps shrink to $\leq 1$ step for most models. Large gaps on *Hard* (e.g., 5.65 steps for Gemini-2.0-FT) versus minimal gaps on *Non-Hard* suggest reasoning depth matters primarily near the model's capability boundary.

### 4.1.2  *RQ1.2 Is verbosity in thinking linked to resolution success?*

*Approach.* While step count measures the breadth of reasoning, verbosity captures its depth. Using the same task-controlled setup, we measure verbosity as the average number of words per step for each attempt. For every model and difficulty split, we compare verbosity on successes vs. failures.

*Results.* The verbosity analysis reveals that the relationship between verbosity and resolution success is model-specific rather than universal, with different models exhibiting distinct preferences that should be understood alongside their step-count behaviors, as discussed in Section 4.1.1.

As shown in Fig. 3, Gemini-2.0-FT succeeds with concise individual steps but utilizes many of them; successful attempts average only 44.27 words per step, compared to 66.59 words on failures (Spearman value $\rho = -0.67$, 95% CI [-0.87, -0.257]). This is complementary to Section 4.1.1, where we showed that Gemini-2.0-FT uses more total steps when succeeding (13.75 vs. 8.10 steps, Spearman value $\rho = 0.79$), indicating that the model breaks down problems into numerous steps while keeping each step focused and efficient. When the model fails, it produces fewer total steps that are individually more verbose, suggesting it becomes stuck elaborating on particular points rather than systematically progressing through the problem. Claude-3.7-Sonnet exhibits a similar, though less pronounced, pattern (43.24 vs. 40.96 words per step), maintaining relatively consistent conciseness across outcomes. Qwen-QwQ shows the inverse pattern on *Hard* tasks: successful attempts are more verbose at 42.44 words per step compared to 29.67 on failures (Spearman value $\rho = 0.71$, 95% CI [0.373, 0.873]). Connecting to section 4.1.1, we found that Qwen-QwQ succeeds with fewer total steps on Hard tasks (4.5 vs. 6.2 steps, Spearman value $\rho = -0.52$), meaning this model compensates for shorter reasoning chains by providing more detailed exploration within each step. When Qwen-QwQ fails on *Hard* tasks, it produces both more steps and less verbose ones, suggesting it engages in shallow exploration rather than focused, deep reasoning. However, this pattern does not hold consistently across difficulty levels. These verbosity–step trade-offs can vary with difficulty; for example, on *Non-Hard* tasks, Qwen-QwQ shows slightly more steps when succeeding (5.06 vs. 4.8) while maintaining higher verbosity (39.23 vs. 33.89 words per step). This indicates that Qwen-QwQ's reasoning strategy adapts to task difficulty; on challenging problems, it succeeds by concentrating detailed reasoning into fewer steps. DeepSeek-R1 and Gemini-2.5-Flash maintain high verbosity regardless of outcome, with both achieving only 14.29% resolution rate on *Hard* tasks. Their high word counts suggest they are over-elaborating without effectively advancing the solution.

> **Answer to RQ1.2** Verbosity interacts with step count in model-specific ways. Some models succeed with many concise steps (Gemini-2.0-FT), while others succeed with fewer but more detailed steps (Qwen-QwQ on *Hard*). Strategies can shift with task difficulty, and high verbosity without strategic progress (DeepSeek-R1, Gemini-2.5-Flash) does not imply success.

## 4.2  Assessing the Reasoning Quality in Thinking Code LLMs (RQ2)

Building on the reasoning analysis in RQ1, we next analyze reasoning quality by examining the reasoning content of the same six thinking LLMs across the same 100 BigCodeBench-Instruct tasks. Using the protocol described previously (Section 3.4), 21 trained annotators evaluate each reasoning chain along three dimensions: **Efficiency**, **Logic Consistency**, and **Completeness**. Figure 4 presents our annotation workflow. We then explore:

**RQ2.1.** What patterns characterize problematic (faulty) reasoning?

**RQ2.2.** How does task complexity (*Hard* vs. *Non-Hard*) affect (i) reasoning quality (efficiency, logic, and completeness) and (ii) solution correctness across models?

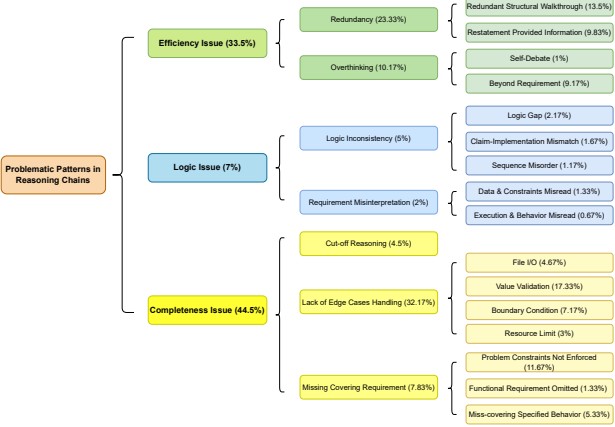

**Figure 5: Taxonomy of the patterns of problematic reasoning patterns. Ratios cannot add up to 100% due to overlapping.**

### 4.2.1 RQ2.1 Patterns in Problematic (faulty) Reasoning

*Approach.* We construct the taxonomy following a rigorous qualitative methodology grounded in open-coding techniques. First, all authors independently examine a subset of failure instances flagged by our 21 human evaluators across the three criteria (Efficiency, Logic Consistency, and Completeness). The authors collaborate in multiple rounds of discussion to consolidate similar patterns. Throughout, we apply constant comparison analysis by continuously contrasting newly observed patterns to ensure distinctiveness and theoretical saturation.

*Results.* Figure 5 presents the taxonomy, where we group problematic reasoning into three top-level categories: **Completeness Issue**, **Efficiency Issue**, and **Logic Consistency Issue** to better map to the three metrics. Each category includes second-level subcategories, and where needed, third-level sub-subcategories reflecting concrete trace manifestations. We define each pattern below and provide representative examples.

**Completeness Issues** are the most frequent failures (44.5%), reflecting systematic gaps in analysis and coverage.

(1) **Cut-off Reasoning:** manifests when models prematurely terminate their reasoning chains, often after identifying surface-level requirements and during the middle of the reasoning chain, but fail to pursue the whole logical progression necessary for comprehensive problem-solving. This pattern typically occurs when models recognize the general problem domain but lack the persistence or capability to work through all necessary reasoning steps, resulting in partial solutions that appear plausible but miss essential components.

(2) **Missing Coverage of Requirements:** the model overlooks explicit constraints, functional requirements, or specified behaviors. We distinguish three subcategories: (a) **Problem Constraints Not Enforced**, where reasoning ignores prompt-stated input assumptions, schema checks, parameter exactness, scope constraints, or required runtime/reproducibility settings that

gate correct execution; (b) **Functional Requirement Omitted**, where a required transformation, algorithm, or computation is missing; and (c) **Miss-covering Specified Behavior**, where outputs violate interface/presentation specs (e.g., returns, files, names, labels, and write options).

(3) **Lack of Edge Case Handling:** the most prevalent and technically significant pattern, revealing weaknesses in robust code generation reasoning. We identify four subcategories: (a) **File I/O**, which encompasses failures to consider file access permissions, non-existent file paths, corrupted file formats, and concurrent file access scenarios that are essential for real-world code reliability; (b) **Value Validation**, which includes insufficient consideration of input sanitation, type checking, format validation, and range verification that are crucial for preventing runtime errors and security vulnerabilities; (c) **Boundary Condition**, includes handling of min/max values, $\{\varnothing, 0, \text{None}\}$ value processing, special numeric values like $\{\text{NaN}, \pm\infty\}$ management that frequently cause production code failures (Fig. 7) ; and (d) **Resource Limits**, which addresses scalability concerns, including large memory allocation scenarios, extensive directory traversal operations that are essential for production-ready code generation.

---

**Problem (BigCodeBench/797)**

Count the total number of brackets across all *values* in a pandas.DataFrame. Brackets are ( ) { } [ ]; use the regex pattern `[(){}[\\]]` to identify them. Non-DataFrame input → raise TypeError. Return an int (e.g., `['(a)', 'b', '[[[[)c']` → 8).

**Model's behavior (observed).** The model vacillates between multiple strategies, fixating on regex escaping minutiae with several competing patterns before eventually settling on `[(){}[\\]]`. The plan gets repeatedly re-stated, delaying a clear, single implementation path.

**Model's Reasoning Steps (summarized):**
- Validate input: Check `isinstance(df, pd.DataFrame)` and raise `TypeError` otherwise.
- Decide traversal: Debates converting the entire `DataFrame` to a string vs. iterating cell-wise; worries about indices/headers leaking; oscillates between options without committing.
- Choose regex: Lengthy back-and-forth on character-class rules and escaping `]` inside `[...]`; proposes multiple variants before concluding with `[(){}[\\]]`.
- Implement counting: Alternates between nested loops, `applymap`+`findall`, and `.str.findall` pipelines; re-raises performance considerations; finally picks `applymap` with a compiled pattern. *Summary:* Repeated re-planning.
- Edge notes: Mentions NaN stringification and empty frames; no further complications.

**Problematic pattern: Circular Reasoning / Overthinking** The reasoning is extremely verbose with excessive detail about strategy selection, regex choice, and implementation method from step 2 to step 4.

---

**Figure 6: Example for Self-Debate of BigCodeBench/797 prompt and brief observation.**

**Efficiency Issues** capture reasoning that may reach correct conclusions but is systematically inefficient and undermines practical reasoning quality.

(1) **Redundancy:** manifests as unnecessarily lengthy reasoning chains that include repetitive explanations, redundant problem restatements, and excessive elaboration that obscures rather than

> **Problem (BigCodeBench/1082)**
>
> Compute the Pearson correlation between numeric `score` and categorical `grade`: (1) convert string scores to floats; (2) encode grades by rank order; (3) return the correlation coefficient as a `float`. If the DataFrame has < 2 rows, return NaN. Starter stub: `import pandas as pd`, `from scipy.stats import pearsonr`, `def task_func(data):`.
>
> **Model's behavior (observed).** Outlines a sensible pipeline and an implementation plan.
>
> **Model's Reasoning Step:** <step1> Input Validation <step2> Convert scores from string to float <step3> Encode categorical grades by rank order <step4> Compute Pearson correlation using scipy <step5> Return correlation coefficient
>
> **Problematic pattern: Lacking Edge Cases Handling** The model did not check for missing columns, non-numeric scores, or dropped NaNs in any reasoning steps.

**Figure 7: Example for lacking edge cases handling: BigCodeBench/1082.**

> **Problem (BigCodeBench/1036)**
>
> Visualize two `Series` with a swarm plot and highlight their intersecting values. The function compares two pandas `Series` in a swarm plot and marks intersections with red dashed lines at the intersecting data values. It returns the plot `Axes` and the number of unique intersections. The solution must be self-contained and start with `import pandas as pd`, `import seaborn as sns`, `import matplotlib.pyplot as plt`, `def task_func(s1, s2):`.
>
> **Model's behavior (observed).** Finds the set intersection of values, reshapes data for `seaborn.swarmplot`, draws dashed lines at intersections, and returns the `Axes` with the intersection count.
>
> **Model's Reasoning Step:**
> <step 1> Clarify goal and recall what a swarm plot shows.
> <step 2> Compute the intersection of the two `Series` using set operations.
> <step 3> Build a two-column DataFrame with value and series label for plotting.
> <step 4> Create the swarm plot to compare the two `Series`.
> <step 5> State that red dashed *vertical* lines will highlight intersections.
> <step 6> Return the `Axes` object and the unique intersection count.
> **Problematic pattern: Logic Inconsistency (Claim vs Implementation Mismatch)**
> The chain specifies vertical dashed lines to mark intersecting values, which mismatches the expected behavior for this visual encoding. In a swarm plot with values on the y-axis, intersection markers should be horizontal at the corresponding y-values; the proposed orientation conflicts with the task intent.

**Figure 8: Example for logic inconsistency: BigCodeBench/1036.**

clarifies the reasoning process. We define two sub-subcategories: (a) **Redundant Structural Walkthrough**, which represents a step that re-describes the plan, structure, or approach the model already established in the previous steps. These steps merely recapitulate existing reasoning organization without advancing the solution; and (b) **Restatement Provided Information**, occurs when the model repeats information already explicitly stated in the problem description or task specification.

(2) **Overthinking:** which captures excessive analysis that does not advance problem solving, appearing as circular deliberation without resolution and expansive reasoning beyond the problem scope. We identify two subcategories: (a) **Self-debate**, which occurs when the model becomes trapped in recursive loops, repeatedly revisiting the same set of alternatives without adding new criteria or making a decision. These circular deliberations consume computational resources and reasoning steps, yet fail

to reach conclusions or advance the solution, representing cognitive spinning that produces no forward progress (Fig. 6); and (b) **Beyond requirement**, represents where the models introduce considerations, constraints, or requirements that are not present in the original problem specification. This pattern often manifests as models fabricating additional complexity, assuming unstated requirements, or addressing problems that extend beyond the specified scope.

**Logic Issues** are the most fundamentally damaging because they break reasoning reliability regardless of other quality factors.

(1) **Logic Inconsistency:** internal contradictions where statements or assumptions that conflict with earlier steps, maintain incompatible definitions or approaches coexist, or draw conclusions that logically contradict their own premises. Although infrequent, these failures are catastrophic. We identify three subcategories: (a) **Logic Gap**, occurs when reasoning steps contain unjustified leaps, missing intermediate inferences, or unsupported assumptions where concepts are introduced or advanced without sufficient logical foundation from the prior step; (b) **Claim-Implementation Mismatch**, represents when claiming one approach but implementing a different or contradictory one later, which the reasoning strategy diverging from the actual code (Fig. 8); and (c) **Sequence Misorder**, represents violations of necessary logical or operational ordering, where the model proposes or reasons about steps in an order that contradicts dependencies, prerequisites, or temporal constraints inherent to the problem.

(2) **Requirement Misinterpretation:** misunderstanding or incorrect parsing of the problem statement. We identify two subcategories: (a) **Execution & Behavior Misread**, when the expected runtime behavior or procedural setup is misunderstood; and (b) **Data & Constraints Misread**, when the data semantics or constraints are misread.

Model-specific trends further highlight distinct failure modes. DeepSeek-R1 is more prone to overthinking, especially self-debate (66.67% of self-debate cases), and self-debate accounts for 8.33% of its problematic traces. Gemini-2.5-Flash most often exhibits cut-off reasoning, including chains cut mid-sentence: it contributes 66.67% of all cut-off cases, and cut-off reasoning accounts for 26.87% of its problematic traces.

> **Answer to RQ2.1** Completeness issues dominate (44.5%), especially missing edge cases. Efficiency issues are secondary (33.5%). DeepSeek-R1 is prone to overthinking/self-debate (8.33% of its failures), and Gemini-2.5-Flash frequently produces cut-off reasoning (26.87% of its failures).

### 4.2.2 RQ2.2 Impact of Task Complexity

*Approach.* To investigate the impact of task complexity on reasoning quality, we conduct two analyses. First, we compute Spearman correlations between reasoning quality metrics and failure rates for the *Hard* and *Non-Hard* task sets to quantify how reasoning quality relates to outcomes across different difficulty levels. Second, we compare pass rates across all six models on *Hard* vs. *Non-Hard* to measure performance degradation and identify which models are most robust to complexity.

**Table 3: Pass rate with Hard vs. Non-Hard set across six models.**

| Model | Pass rate (%) | |
| --- | --- | --- |
| | **Hard** | **Non-Hard** |
| Gemini-2.5-Flash | 14.29% | 36.05% |
| O3-mini | 35.70% | 55.81% |
| Gemini-2.0-FT | 28.57% | 41.86% |
| DeepSeek-R1 | 14.29% | 37.21% |
| Claude-3.7-Sonnet | 21.43% | 29.07% |
| Qwen-QwQ | 28.57% | 39.53% |

*Results.* Our analysis reveals a negative correlation between task complexity and reasoning quality, with completeness emerging as the most predictive metric of task outcomes. In the *Hard* task set, completeness demonstrates a small but statistically significant negative correlation with failure rates (Spearman r = -0.219), indicating that as reasoning completeness decreases, failure probability increases substantially. This relationship, while still present in the *Non-Hard* task set, is considerably weaker (r = -0.096), suggesting that task complexity amplifies the importance of complete reasoning. The differential correlation strength between *Hard* and *Non-hard* sets provides compelling evidence that reasoning quality becomes increasingly critical as task difficulty escalates, with incomplete reasoning serving as a more pronounced predictor of failure in challenging scenarios.

The solution correctness results show substantial performance variances both across models and complexity levels, as shown in Table 3. O3-mini demonstrates superiority on both *Hard* (35.70%) and *Non-Hard* (55.81%) sets, followed by a middle tier of models, including Gemini-2.0-Flash-Thinking, Qwen-QwQ, and DeepSeek-R1 on the *Non-Hard* dataset, while Gemini-2.5 and DeepSeek-R1 exhibit notably poor performance (both at 14.29%). The complexity-included performance degradation varies significantly across models, with DeepSeek-R1 and o3-mini experiencing the largest drops (22.92% and 21.30% points, respectively) when transitioning from *Non-Hard* to *Hard* tasks, while Claude-3.7-Thinking shows remarkable resilience with only a 7.64% point decrease, suggesting that different models have different capabilities when handling complex problems. In addition, the large drop of o3-mini suggests that top performance does not confer hardness robustness, while Claude-3.7-Thinking shows the most stable performance.

> **Answer to RQ2.2** Task complexity degrades both reasoning quality and outcomes. Completeness is more predictive on *Hard* tasks ($\rho = -0.219$) than on *Non-Hard* tasks ($\rho = -0.096$), showing that incomplete reasoning becomes a stronger failure indicator as difficulty increases.

## 5 Recommendations from Lessons Learned

Table 4 suggests several methodological implications for studying reasoning traces in code LLMs, clarifying which properties of reasoning traces are robust signals versus model- and difficulty-dependent artifacts that require careful experimental control.

**Report reasoning structure with difficulty stratification:** Because the step-success relationship varies across models and diverges most on more difficult tasks, evaluations on thinking code LLMs should report reasoning structure statistics separately, such as step count and verbosity, across difficulty levels when such labels exist in the benchmark.

**Evaluate beyond pass rates with process diagnostics and failure taxonomies:** Outcome-only metrics (e.g., pass@k) hide why a model failed and whether the trace is useful for debugging. We recommend that future evaluations report focused, process-oriented signals alongside correctness, at minimum: the coverage of the reasoning chain for requirements (identify all stated requirements from the problem and mark each as addressed/not addressed), constraints (verify that input assumptions, parameter exactness, and scope limits are acknowledged), and edge cases (check for file I/O handling, value validation, boundary conditions, and resource limits); the logic and step-by-step coherence (judge whether each step logically follows from prior steps and mark as coherent/incoherent); and the meaningfulness (label each step as essential or non-essential based on whether it advances the solution) and usefulness (identify which steps introduce necessary information or perform required calculations) of the reasoning traces.

**Improve coding by making coverage adaptive, not longer:** Since task complexity amplifies the impact of incomplete reasoning (as evidenced by stronger negative correlation between completeness and failure on *Hard* tasks, $\rho = -0.219$, vs. *Non-Hard* tasks, $\rho = -0.096$), future thinking code LLMs should move from "generate longer traces" to an adaptive coverage policy: start with a short plan and a concrete checklist (requirements, constraints, edge cases), expand reasoning only while new checklist items remain unresolved, and stop once coverage is complete.

**Table 4: Takeaway messages and mapped RQ answers.**

| Takeaway messages | Mapped ans. |
| --- | --- |
| Longer chains are not universally better; report step-count statistics and step–success associations separately by difficulty. | ans. RQ1.1 |
| Avoid using verbosity alone as a correctness proxy; interpret verbosity jointly with step count (strategy trade-offs). | ans. RQ1.2 |
| Evaluate beyond pass rates using process diagnostics and a failure-pattern breakdown to make errors actionable. | ans. RQ2.1 |
| Improve models via adaptive coverage, rather than simply generating longer traces. | ans. RQ2.2 |

## 6 Threats to Validity

Our results may not generalize broadly because we evaluate only BigCodeBench and the Python language and a stratified 100-task sample, which may miss rare tail cases. However, BigCodeBench provides broad real-world coverage (1,140 tasks across 139 libraries and seven domains). Our sample preserves its Hard/non-Hard distribution, so our main cross-model and difficulty-stratified trends are unlikely to depend on outliers. Human ratings from 21 CS graduate students and our efficiency-logic-completeness framework raise threats to generalizability. We mitigate this by training the coders with a detailed codebook and by ensuring all students have working backgrounds in ML/LLM. The "reasoning quality" is partly subjective because judging step necessity and implicit requirements involves human judgment. We mitigate this with a detailed rubric, training, three independent ratings per task, and strong inter-rater

agreement. Some models required retries only when an output was empty. We mitigated this by reusing the same request and the first complete output. We treated "successful" outputs as those that pass the benchmark tests. Finally, temporal validity is limited because LLM behavior changes rapidly. We report model versions and dates and interpret our findings as a time-stamped snapshot.

## 7 Conclusion

We empirically evaluated six thinking LLMs on 100 diverse Big-CodeBench code-generation tasks. We find that reasoning-chain length has a non-monotonic, model-specific relationship with success. Failure analysis shows that completeness is the main bottleneck (44.5%) followed by efficiency (33.5%) and logical inconsistencies issues (7.5%). We observed that the major root causes behind incomplete reasoning was missing edge cases/boundary conditions. Incompleteness correlates more strongly with failure on *Hard* tasks ($\rho = -0.219$) than on *Non-Hard* tasks ($\rho = -0.096$). Overall, these results highlight that thinking models often under-specify requirements and implicit constraints. We observed that longer chains or larger inference budgets do not reliably fix this. The results point to the need for better requirement assessment, edge-case awareness, and more strategic allocation of reasoning efforts.

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
