# OpenReview forum: "An Empirical Study of Reasoning Steps in Thinking Code LLMs"
_ACM.org/AIWare/2026/Conference — AIware 2026_

### Official Review · Reviewer_C7PR · 2026-03-08

**Rating:** 3
**Confidence:** 4

**Review:**

This paper studies the reasoning traces of six thinking code LLMs on 100 BigCodeBench tasks and analyzes them along two main axes: structural properties of the traces, such as step count and verbosity, and human-judged reasoning quality, measured through efficiency, logical consistency, and completeness. It further develops a taxonomy of problematic reasoning patterns from the manually inspected traces. Overall, I found the paper clear, timely, and methodologically thoughtful in several respects. These are the strengths that I found:

1. **Relevant and underexplored research direction**: The paper addresses an important question that is still relatively underexplored in code LLM evaluation: not just whether models solve programming tasks, but what their explicit reasoning traces look like and how reasoning quality relates to correctness. This is a valuable direction because most prior evaluations emphasize outcome metrics such as pass@k or test success, which reveal little about where or how reasoning breaks down. The paper’s focus on process-level analysis fits well with the broader goal of making reasoning models more interpretable and diagnosable.
2. **Rigorous human evaluation setup**: A major strength is the human evaluation protocol. The study uses 21 participants, applies a detailed codebook, assigns three independent evaluators per task, and reports strong inter-rater reliability using ICC(1,3) across all three dimensions. This gives the human analysis substantially more credibility than many papers that rely on informal qualitative inspection. The efficiency, logical consistency, and completeness framework is also intuitive and appropriate for reasoning-trace assessment.
3. **Useful taxonomy of problematic reasoning patterns**: The taxonomy of failure patterns appears to be the most novel and reusable contribution of the paper. In particular, the decomposition of completeness failures into concrete subcategories such as missing value validation, missed problem constraints, and missed boundary conditions makes the analysis more actionable than a coarse success/failure breakdown. This taxonomy could be useful for future work on reasoning-trace diagnosis, training data construction, or targeted interventions.

I also found some areas that this paper can be improved:

1. **Very small Hard-task sample weakens several key claims**: The most important limitation is that the study includes only 14 Hard tasks in its 100-task sample. Yet a number of the more interesting claims rely on correlations computed on this small subset. With at most 14 observations per model, those correlations are highly sensitive to individual tasks and outliers. The paper does not report confidence intervals, significance tests, or robustness checks such as leave-one-out analysis. As a result, the Hard-task findings are better interpreted as exploratory than as strong evidence of stable model differences.

2. **Prompt standardization may itself shape the observed reasoning traces**: The paper uses a structured prompt that reformats reasoning into clearly delimited steps across all models. This is understandable from a comparability standpoint, but it also introduces a potential confound: measured step counts and reasoning verbosity may partly reflect the elicitation protocol rather than the models’ natural reasoning behavior. The paper acknowledges that results are prompt-conditional, but the implications of this design choice deserve more discussion, especially because step count is central to the paper’s analyses and conclusions.

3. **Lack of non-thinking or conventional baseline models limits comparative insight**: The paper evaluates only thinking LLMs. This is reasonable given its scope, but it also limits the strength of its broader implications. Without including non-thinking models, or at least chain-of-thought-prompted conventional code LLMs, it is difficult to tell whether the identified reasoning failures are distinctive to thinking models or simply common code-generation failure modes expressed in more verbose form.

4. **The study does not examine reasoning budget as an experimental factor**: The paper is fundamentally about reasoning depth and step structure, yet it evaluates each model under only one fixed prompting/configuration regime. That choice is defensible for a controlled comparison, and the paper explicitly frames the results as prompt-conditional rather than leaderboard-optimized. Still, given the paper’s central questions, it is a notable limitation not to explore how controllable reasoning effort or budget affects the observed traces and conclusions.

**Minor comments**:
1. Table 2 reports total step counts by model, but a Hard versus Non-Hard breakdown would better support the RQ1 analysis.
2. The sampling justification based on 95% confidence and roughly 9 to 10% margin of error is appropriate for estimating proportions over the benchmark, but it does not directly justify the precision of correlation analyses or the adequacy of the qualitative taxonomy.
3. The taxonomy percentages are said to overlap, but the paper does not clearly explain how overlapping cases were counted, which makes the reported percentages harder to interpret.
4. Models are old. It would be good to include at least one recent model.

**Summary:**

This paper presents an empirical study of reasoning traces produced by six “thinking” LLMs (DeepSeek R1, OpenAI o3 mini, Claude 3.7 Sonnet, Gemini 2.0 Flash Thinking, Gemini 2.5 Flash, and Qwen QwQ) on 100 BigCodeBench code generation tasks. The authors analyze 600 model-task runs containing 3,772 reasoning steps. Two research questions investigate (RQ1) whether reasoning structure (step count and per-step verbosity) relates to solution correctness and (RQ2) what reasoning failures characterize incorrect solutions. A human evaluation with 21 participants assesses reasoning quality across efficiency, logical consistency, and completeness. Results show that the relationship between reasoning length and success is model- and difficulty-dependent, verbosity alone is not a reliable correctness signal, and reasoning failures are dominated by completeness issues (44.5%), particularly missing edge-case handling. The study further develops a taxonomy of problematic reasoning patterns across efficiency, logic, and completeness dimensions.

---

> ### Author Response · Authors · 2026-03-21
> **Response to Reviewer C7PR**
>
> We sincerely appreciate the reviewer's detailed comments. We address all questions below.
> 1. **Very small Hard-task sample.**  We appreciate the comment. We would like to clarify that our analysis wants to show that near the capability limit, the step–success relationship becomes model-dependent rather than uniform. BigCodeBench contains 148 Hard tasks out of 1140 total tasks (13.2%), and our stratified sample closely preserves that distribution, with 14 Hard and 86 Non-Hard tasks. We here report the 95% CI on Hard tasks, Gemini-2.0-Flash-Thinking succeeds with 13.75 steps on average versus 8.10 on failures and shows a positive step–success correlation (𝜌  = 0.79, 95% CI[0.46, 0.89]), whereas DeepSeek-R1 (𝜌 = -0.58, 95% CI[-0.84, -0.38]) and Qwen-QwQ (𝜌  = -0.52, 95% CI[-0.83, -0.14]) show the opposite pattern, succeeding with fewer steps. In contrast, on the Non-Hard split, correlations collapse toward zero, and the success–failure step gaps shrink to at most about one step for most models. Given the wide 95% CIs, we clarify that the Hard-split results are not conclusive and leave deeper investigation for future work.
>
> 2. **Prompt standardization may itself shape the observed reasoning traces.** We appreciate this comment. We used a standardized prompt to enable a controlled cross-model comparison of visible reasoning traces, because the six studied models expose reasoning in different formats. For example, DeepSeek-R1 provides a separate response for thinking, whereas Gemini-2.0-FT and Qwen-QwQ interleave reasoning with the final answer in a single output. The differences in step count, verbosity, and reasoning structure would be strongly shaped by model-specific presentation styles, making cross-model analysis less interpretable. This motivated our use of a shared step-delimited format to support reliable analysis and human evaluation. We therefore standardized the prompt across all models and interpreted the findings as prompt-conditional under a common protocol, which we believe provides the fairest basis for comparison. At the same time, we recognize that no single prompting strategy can fully eliminate internal-validity risk in cross-model comparisons, because each model has its own prompt-response characteristics. We want to mitigate this threat as much as possible by applying the same prompting and evaluation protocol to every model in a consistent and controlled setting. We agree that model-specific prompt behavior is important in its own right and see this as valuable future work.
>
> 3. **Lack of non-thinking or conventional baseline models.** We appreciate this thoughtful suggestion. In this paper, we focus on thinking LLMs because our analysis centers on the quality of externally visible reasoning traces. This makes reasoning-enabled models a natural starting point, as they provide intermediate traces that can be directly examined and audited. We agree that including non-thinking or conventional code LLMs would provide valuable insight into whether the identified failure patterns are specific to thinking models or reflect broader code-generation challenges. We therefore view such comparisons as an important future work. Extending the study to conventional models, including chain-of-thought-prompted variants where appropriate, could clarify which issues are unique to explicit thinking traces and which are shared across code LLMs. We thank the reviewer and will make this future direction clearer in the paper.
>
> 4. **The study does not examine the reasoning budget as an experimental factor.** We appreciate this concern and suggestion. We did not include this part of the experiment due to the page limits and focus of our current research questions. We have existing control evidence on this point. In o3-mini, low/medium/high reasoning effort produces only modest shifts in reasoning quality and final pass rates, with medium performing best in our observed setting (Hard 35.71%, Full 56.98%). Separately, our step-budget manipulations show that modest 10–30% cuts often preserve success on Full tasks but rarely on Hard tasks, and that increasing steps does not yield uniform gains. These controls suggest that reasoning effort matters, but not monotonically: more budget alone is not a universal remedy.
>
> 5. **Minor comments** We appreciate these comments and the opportunity to clarify. The Hard vs. Non-Hard split breakdown is already reflected in the step and verbosity analyses in Figure 2 and 3. On the taxonomy percentages, it is correct to read as overlapping, non-exclusive frequencies: a single faulty trace can simultaneously exhibit, e.g., missing edge-case handling and redundant structural walkthrough. This is why the totals do not sum to 100%. Finally, although the model landscape is evolving quickly, we position the paper as a controlled comparative study whose main value lies in difficulty-stratified trace analysis, human evaluation, and a failure-pattern taxonomy.

---

### Official Review · Reviewer_RHJ2 · 2026-03-10

**Rating:** 4
**Confidence:** 4

**Review:**

Overall the paper is well written, well motivated and provides useful empirical insights in reasoning LLM failures.

There is an unacknowledged a threat to validity for RQ1: faithfulness of self-reported reasoning trace breakdown to actual steps under the custom prompt for the trace-formatting protocol. The prompt is not included in the paper and if the native reasoning traces are substantially different, the prompt is not just a hyperparameter but may crucially affect statistics for RQ1.
This can be relatively easily double-checked and demonstrated, at least for some models.

## Feedback
The paper would also benefit from
* reporting the actual prompt for eliciting reasoning step breakdown (e.g. in an appendix). Although it seems available in the replication package, it is copy-pasted as a string literal among 6 per-model .py files, which makes it hard to validate e.g. if it is the same for every model, as the paper claims

* sharing the labeling guide/protocol used by the 21 graduate student participants for annotating reasoning traces (e.g. as part of the replication package)

* sharing intermediate results of labeling tasks for Completeness from "3.4 Evaluation Methodology" as a dataset
  > Participants identify all essential requirements in the problem statement, then mark each as “Addressed” or “Not addressed” in the reasoning.
  That would increase transparance of the qualitative evaluation as well as foster further explorations in reasoning failure modes.

* more care around reporting output parsers e.g. in "3.3 Data Preparation"
  > For example, DeepSeek-R1 provides a built-in separation between the reasoning process and the final response (e.g., "content" vs. "reasoning_content")

  from experience, one may guess that the authors probably mean the [reasoning output parsers in vLLM's OpenAI-compatible API server](https://docs.vllm.ai/en/v0.8.4/features/reasoning_outputs.html), but this is not clear from the text and would be a usefull information for practitioners

* a one-sentence BigCodeBench dataset refresher (is it function-level? Where does execution feedback come from?)


## Questions
The [replication package](https://github.com/xhinini/Reasoning-LLMs/tree/main?tab=readme-ov-file#quick-start) appears somehow incomplete:
1. instructions mention `eval/evaluate.py` but there is no such file; are the instructions out of date?
2. It includes the code for automated LLM-as-a-Judge labeling of the reasoning traces across the same 3 dimensions that seems not to be reported in the paper (it only mentions 21 participants doing that). Why? From practical standpoint it would be interesting to see whether it agrees with human judgment.

**Summary:**

The authors run 6 thinking LLMs over 100 BigCodeBench-Instruct difficultiy-stratified tasks of humanEval-like function-level code generation and
* Quantitatively study resoning traces lengh corelation with execution-based feedback
* Qualitatively study a subset of 10 examples manually labeled by 21 human annotators (graduate students) across Efficiency/Coherence/Completeness dimensions and propose a taxonomy of failures

Central practical finding: longer reasoning chains and higher inference budgets don't fix the core failure mode (the models systematically under-specify edge cases).

---

> ### Author Response · Authors · 2026-03-21
> **Response to Reviewer RHJ2**
>
> We sincerely appreciate the reviewer's detailed comments. We address all questions below.
> 1. **On the faithfulness threat for RQ1 and the role of the trace-formatting prompt.** We appreciate this comment. To improve transparency, we provide the exact elicitation prompt used in our experiments below. Our objective in using this prompt was to ensure a fair and consistent comparison across models by standardizing how reasoning traces were presented, since different providers expose reasoning in different output formats. The prompt was designed to normalize the organization of the reasoning trace into clearly delimited steps, rather than to introduce model-specific guidance or unequal scaffolding. We therefore report our findings as controlled, prompt-conditional comparisons under a shared elicitation protocol.
> Problem Description
> {problem_description}
>
> Please carefully study this software engineering problem, conduct a comprehensive analysis, and provide a solution with your reasoning process.
>
> As an expert software developer, your task is to:
>
> Understand the requirements
> Design an approach to solve the problem
> Implement the solution in code
> Verify the correctness of your solution
>
> Format to Follow:
> Reasoning Process:
> [Please explain your thinking process step by step, with each logical step in a separate paragraph, and use a format such as <step 1> to label each step.]
>
> <step 1> Specific thinking content of this step <step 2> Specific thinking content of this step ... Specific thinking content of this step
> Solution:
> [Provide your complete code implementation here. Ensure it is functional, efficient, and addresses all requirements.] """
>
> 2. **Sharing the labeling guide/protocol & results of labeling tasks.** We appreciate the suggestion. We have made the labeling guide and evaluation results available in our replication package.
>
> 3. **More care around reporting output parsers e.g. in "3.3 Data Preparation".** We appreciate this comment. We also describe the relevant distinction in the paper like, DeepSeek-R1 exposes “content” vs. “reasoning_content”, whereas models such as Gemini-2.0-Flash-Thinking and Qwen-QwQ may combine reasoning and final answer in a single output. That heterogeneity is exactly why we used one shared standardized output format for the evaluation. We agree that this clarification is useful for practitioners because it explains why normalization was methodologically necessary, and we will make it clearer in the revision.
>
> 4. **A one-sentence BigCodeBench dataset refresher (is it function-level? Where does execution feedback come from?).** We appreciate for this helpful suggestion. BigCodeBench [1] is a function-level code-generation benchmark, and each example asks the model to generate a single target function and is evaluated using associated unittest-style test code. The benchmark contains 1,140 tasks, with about 5.6 test cases per task and roughly 99% average coverage. In our study, pass/fail is determined by executing generated code through the benchmark’s evaluator backends. We will add a one-sentence BigCodeBench dataset description in our revision to better clarify the task.
> [1] https://github.com/bigcode-project/bigcodebench
>
> 5. **Instructions mention eval/evaluate.py, but there is no such file; are the instructions out of date? It includes the code for automated LLM-as-a-Judge labeling of the reasoning traces across the same 3 dimensions that seem not to be reported in the paper (it only mentions 21 participants doing that). Why? From a practical standpoint, it would be interesting to see whether it agrees with human judgment.**  We thank the reviewer for carefully examining the artifact and for pointing out this source of confusion. The eval/evaluate.py file and the associated LLM-as-a-Judge material were obsolete leftovers from an earlier exploratory study and were not used in any part of the current paper. All results reported in this submission are based on our human evaluation protocol, which we consider the more trustworthy and appropriate basis for this study’s reasoning-quality assessment. To avoid further misunderstanding, we have already removed these files from the repository and revised the corresponding repository instructions with a clearer description. We sincerely appreciate the reviewer for catching this issue, and we apologize for any confusion it caused.

---

### Official Review · Reviewer_qLiF · 2026-03-11

**Rating:** 3
**Confidence:** 3

**Review:**

#### Pros:
- Good analysis paradigm, i.e., shifting from outcome-only pass@k evaluation to examining reasoning trace quality
- Systematic failure taxonomy for problematic reasoning patterns
- Well-designed human evaluation with 21 trained annotators and three independent ratings per task
- Replication package is available
---
#### Cons (kindly see my detailed comments below):
- The sample consists of only 100 tasks, and the `Hard' split contains only 14, which seems too few to support the statistical claims
- The unified structured prompt may introduce bias in the results
- The study could include more recent models, as well as some causal analysis to empirically validate the recommendations in Section 5
---
#### Detailed Comments
Overall, I appreciate the paper as it focuses on reasoning process quality rather than just final correctness, which I believe is a timely and practically relevant research topic. The taxonomy of problematic patterns is a genuine contribution that future work can build on. That said, I have several concerns:

The primary one is the statistical foundation of the difficulty-stratified analysis. As with only 14 `Hard' tasks, the reported Spearman correlations are very likely unstable. The authors could at least report confidence intervals for all Hard-split correlations, or ideally -- expand the Hard sample to provide more reliable estimates. Otherwise, the core claim on reasoning depth is not convincingly supported.

Second, the use of a standardized structured prompt to "present reasoning in clearly delimited steps" is a reasonable choice but introduces a potential confound: different models may respond to the formatting template differently, causing them to output more or fewer steps than their native reasoning would produce. A brief comparison between the prompted format and a model's native output (like DeepSeek-R1's built-in reasoning traces) would help disentangle prompt effects from genuine reasoning behavior.

In addition, some minor points: several of the studied models (o3-mini, Gemini-2.0-FT) date from early 2025, and more recent reasoning models may exhibit substantially different behaviors. The authors could discuss what this means for the generalizability of their findings to the current model landscape. The study would also benefit from some causal analysis. Like, does prompting models with an explicit edge-case checklist, derived from the authors' own taxonomy, improve completeness scores and pass rates? Such an experiment would transform the recommendations in Section 5 to empirically grounded guidance.

**Summary:**

This paper presents an empirical study examining the reasoning processes of six state-of-the-art reasoning LLMs (DeepSeek-R1, OpenAI-o3-mini, Claude-3.7-Sonnet, Gemini-2.0-Flash-Thinking, Gemini-2.5-Flash, and Qwen-QwQ) on 100 BigCodeBench code generation tasks (600 model–task instances; 3,772 reasoning steps).. Through quantitative analysis of 3,772 reasoning steps and a 21-participant human evaluation, the study investigates whether step count and verbosity are related to correctness, and what problematic reasoning patterns lead to failures. Key findings include that the relationship between step count and success is model- and difficulty-dependent, that verbosity is not a reliable proxy for correctness, and that completeness issues (44.5%) dominate failures, primarily due to missing edge-case handling.

---

> ### Author Response · Authors · 2026-03-21
> **Response to Reviewer qLiF**
>
> We sincerely appreciate the reviewer's detailed comments. We address all questions below.
> 1. **The sample consists of only 100 tasks, and the Hard split contains only 14, which seems too few to support the statistical claims.**  We appreciate this comment. Our 100-task sample was drawn to preserve the original BigCodeBench dataset difficulty distribution, so the resulting 14 Hard and 86 Non-Hard tasks closely match the benchmark’s underlying composition (13.16% Hard and 86.84% Non-Hard). And we show that, near the capability limit, the step–success relationship becomes model-dependent rather than uniform. To address the concern about the small Hard split, we report 95% bootstrap confidence intervals for the Hard-split Spearman correlations. In the step–success analysis, Gemini-2.0-FT shows a relatively strong positive association (𝜌 = 0.79, 95% CI [0.461, 0.891]), showing that more steps are associated with higher success on Hard tasks; the CI is fully above 0, supporting a positive association. In contrast, DeepSeek-R1 (𝜌  = -0.579, 95% CI [-0.837, -0.381]) and Qwen-QwQ (𝜌  = -0.523, 95% CI [-0.826, -0.143]) show negative correlations, indicating higher success with shorter chains. In the verbosity–success analysis, Gemini-2.0-FT shows a negative association (𝜌  = -0.67, 95% CI [-0.87, -0.257]), indicating that it succeeds with fewer verbosity steps; combined with the step–success result, this suggests success with many concise steps. In contrast, Qwen-QwQ shows a positive correlation (𝜌  = 0.71, 95% CI [0.373, 0.873]), indicating that higher success is associated with more verbosity steps; combined with the step–success result, this suggests success with fewer but more detailed steps. Given the wide 95% CIs, we clarify that the Hard-split results are not conclusive, and we leave deeper investigation to future work. This broader pattern is also triangulated by signals beyond the coefficients alone. The success–failure step gap diverges much more on Hard tasks than on Non-Hard tasks. For example, for Gemini-2.0, on the Hard split, successful tasks average 13.75 steps while failed tasks average 8.11 steps, a gap of 5.64. On the Non-Hard split, successful tasks average 8.11 steps and failed tasks average 7.84 steps, a gap of only 0.27. Thus, on Hard tasks, success and failure are separated by 5.64 steps, whereas on Non-Hard tasks, they are separated by only 0.27 steps.
>
> 2. **The unified structured prompt may introduce bias in the results.**  We appreciate this comment. The standardized prompt was used to ensure cross-model comparability, as without a unified elicitation format, differences in model-specific output styles would make step-count and reasoning content analysis much less interpretable. From our experience, we ran DeepSeek-R1 first without a standardized prompt, and it produced long, hard-to-read content. To improve the interpretability, we used the standardized prompt across all models.
> At the same time, we agree that the standardized format may introduce some prompt effect, as different models can respond to the same template in different ways. More broadly, there is currently no single method that can eliminate this concern while still preserving fair cross-model comparison [1]. Our Unified prompt was a best-effort design decision to balance interpretability and comparability across models. We will clarify this limitation more explicitly in the paper and consider more targeted comparisons between prompted structured outputs and models’ native reasoning traces as an important future work.
>
> [1] Sclar, M., Choi, Y., Tsvetkov, Y., & Suhr, A. (2023). Quantifying Language Models' Sensitivity to Spurious Features in Prompt Design or: How I learned to start worrying about prompt formatting. arXiv preprint arXiv:2310.11324.
>
> 3. **The study could include more recent models, as well as some causal analysis to empirically validate the recommendations in Section 5.**
> We appreciate this suggestion. We agree that the reasoning-model landscape is evolving quickly, and repeating the study on newer models and adding intervention-based causal analysis would both be valuable next steps. Our model set was chosen to provide a cross-provider snapshot of publicly available reasoning code models at the time of data collection, spanning both open and closed models. Accordingly, we position the paper primarily as a controlled comparative study of reasoning-trace behavior and failure patterns under a shared evaluation setting. While newer models may differ in absolute performance, we believe the paper’s main value extends beyond specific model sets through its difficulty-stratified trace analysis, human evaluation of reasoning quality, and taxonomy of failure patterns. We also appreciate the suggestion of testing an explicit edge-case checklist derived from the taxonomy, and agree this would be a meaningful way to assess its practical utility; we leave this to future work.